

# Evaluating the therapeutic effect of tumor treating fields (TTFields) by monitoring the impedance across TTFields electrode arrays

Xing Li[1], Moshe Oziel[2] and Boris Rubinsky[3]

[1] National University of Defense Technology, Electronic Countermeasure Institute, Hefei, Anhui, China
[2] Department of Physiology and Pharmacology, Tel Aviv University, Tel Aviv, Israel
[3] Department of Mechanical Engineering, Department of Bioenegineering, University of California, Berkeley, CA, USA

## ABSTRACT

**Background**. Tumor Treating Fields (TTFields), are a novel, non-invasive tissue ablation technology for treatment of cancer. Tissue ablation is achieved through the continuous delivery of a narrow range of electromagnetic fields across a tumor, for a period of months. TTFields are designed to affect only cells that divide and to interfere with the cell division process. The therapy is monitored with MRI imaging, performed every couple of months. Current technology is unable to assess the treatment effectiveness in real time.

**Methods**. We propose that the effect of the treatment can be assessed, in real time, by continuously measuring the change in electrical impedance across the TTFields delivery electrode arrays. An *in vitro* anatomic skull experimental study, with brain and tumor mimics phantom tissues was conducted to confirm the potential value of the proposed monitoring system.

**Results**. Experiments show that measuring the change in the impedance amplitude between opposite TTFields electrode arrays, at a typical TTFields treatment frequency of (200 kHz), can detect changes in the tumor radius with a sensitivity that increases with the radius of the tumor. The study shows that TTFields electrode arrays can be used to assess the effectiveness of TTFields treatment on changes in the tumor dimensions in real time, throughout the treatement. This monitoring system may become a valuable addition to the TTFields cancer treatment technology. It could provide the means to continuously assess the effectiveness of the treatment, and thereby optimize the design of the treatment protocol.

# INTRODUCTION

Tumor Treating Fields (TTFields) are a minimally invasive, non-contact, tissue ablation technology that employs intermediate-frequency (100–300 kHz), and low-intensity (<3 V/cm) electric fields to inhibit the growth of dividing (cancer) cells

Corresponding author
Boris Rubinsky,
rubinsky@berkeley.edu

(*Kirson et al., 2004*). Clinical evidence shows that the delivery of TTFields prolongs the survival time of glioblastoma multiform (GBM) patients, without obvious side effects (*Guberina et al., 2020*; *Mun et al., 2018*). The Food and Drug Administration (FDA) has approved the use of TTFields for GBM treatment (*Guberina et al., 2020*; *Mun et al., 2018*). In clinical practice, TTFields are delivered in paired orthogonal directions, left–right and anterior-posterior using insulated ceramic disk electrode arrays, attached to a patient's shaved scalp (*Trusheim et al., 2017*). A schematic of the placement of the electrodes in relation to the tumor, is shown in Fig. 1A. The electrodes are rigidly attached to the shaved scalp and are connected to a portable power generator. To maximize the intensity of the electric fields delivered to the tumor, the placement of the treatment electrodes is individualized to each patient, using the patient own baseline magnetic resonance image (MRI) of the brain to determine the optimal placement. The NovoTAL System (NovoTAL, USA) is a commercial software for optimization of the TTFields electrode placements. A detailed description of the way in which the electrode placements and the choice of treatment parameters are optimized is given in *Trusheim et al. (2017)*. TTFields treatment is substantially different from other clinical tissue ablation treatments. In most conventional tissue ablation treatments, such as microwave, radiofrequency, cryosurgery, focused ultrasound, irreversible electroporation, radiation therapy, the surgical procedure is minutes long. The ablative energy is delivered under real time medical imaging. The success of these treatments can be assessed shortly after the end of the procedure, with conventional medical imaging. Unlike other tissue ablation treatments, which affect all the cells in the treated volume, TTFields affect only the tumor cells that divide. The TTFields electromagnetic fields interfere with the cell division process and stop the cell division (*Li, Yang & Rubinsky, 2020*). Therefore, TTFields are delivered for long periods to affect cells that divide, whenever they are dividing. In conventional TTFields treatment, the electric fields are delivered for many months and even years (*Davies, Weinberg & Palti, 2013*). The patient activates a portable power supply to deliver the treatment. The treatment usually lasts for up to 18 h per day (*Mrugala et al., 2014*). The effect of TTFields is frequency depended and the optimal frequency can vary with cell type (*Giladi et al., 2015*). For GBM, the optimal frequency is thought to be about 200 kHz. For technical and economic reasons the success of the treatment on GBM is assessed by a limited number of follow-up MRI scans, which are usually performed every two months (*Trusheim et al., 2017*; *Hottinger, Pacheco & Stupp, 2016*). If these once every two months MRI scans show that the treatment is not successful, the remedy is to change the treatment parameters. However, GBM is an aggressive disease in which the tumor can grow fast. A technology for real time monitoring of the tumor growth or recession during the actual TTFields treatment could provide timely feedback to the physician on the efficacy of the treatment parameters.

Typical TTFields treatment protocols are as follows. After clinical verification of the nature of the tumor, MRI is used to precisely determine the location of the tumor and its dimensions. Then, the MRI images are used with an optimization algorithm, the NovoTAL System (*Trusheim et al., 2017*) (NovoTAL, USA) to calculate the optimal placement of the treatment electrodes, in such a way that the resulting electromagnetic fields are focused onto the tumors, throughout the months long application of the treatment. In this paper
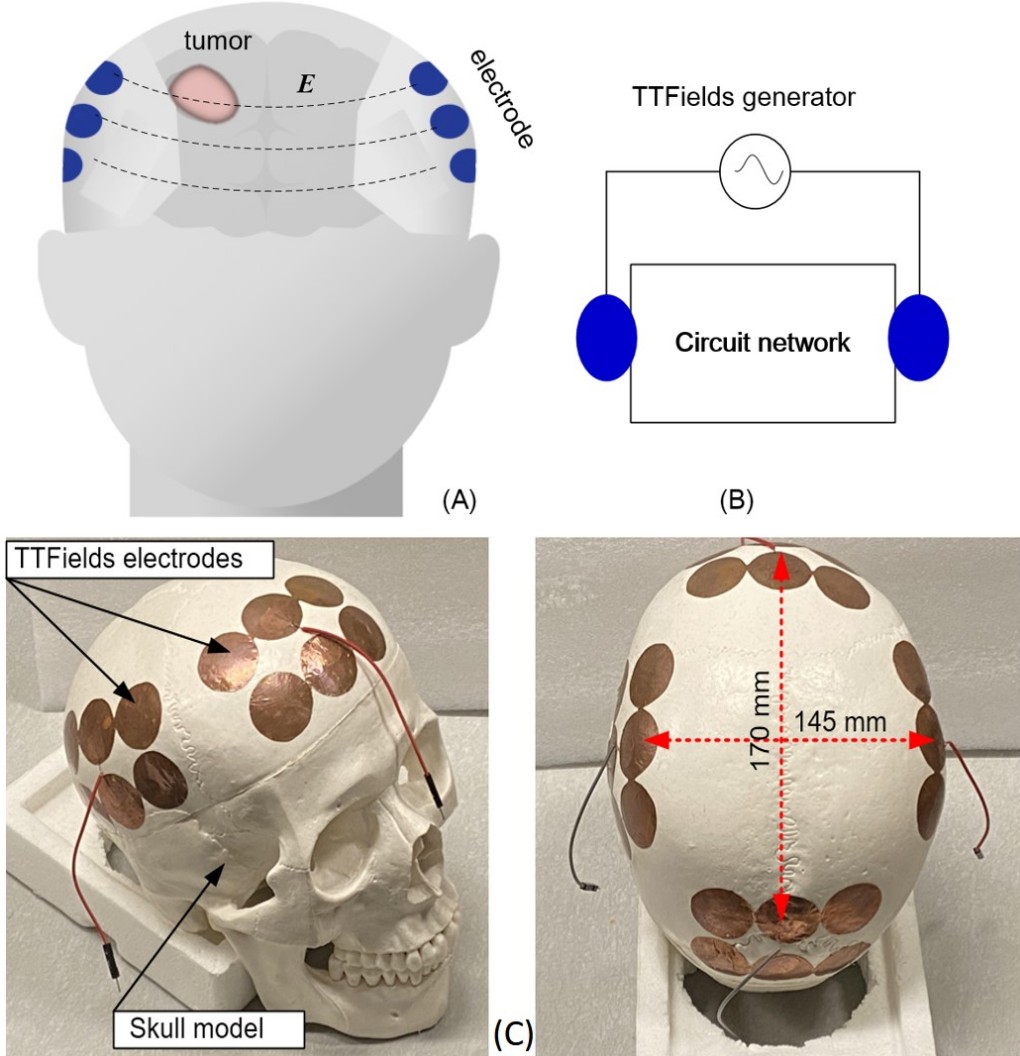

**Figure 1 Schematics and photographs TTFields treatment of TTFields electrode arrays.** TTFields treatment of GBM: (A) schematic of the configuration of electrode array on the skull, (B) equivalent lump model circuit, (C) photographs of the anatomical skull and the TTFields electrode arrays.

we show that the way in which TTFields are delivered across tumors, with electrodes fixed to the skull at a location calculated to deliver the electric fields precisely to the tumors, makes the treatment electrodes particularly suitable for real time monitoring of tumor size through the measurement of the electrical impedance between the electrodes. Changes in the tumor size will affect the electrical impedance between electrodes because normal and malignant brain tissues have different electric properties (*Latikka & Eskola, 2019*).

The head can be viewed as a complex electric circuit network, represented by a black box. The electrode arrays on the scalp are the accessible nodes to the black box network, which is the head, as shown in a schematic way in Fig. 1B. Because of the difference in electrical properties between the normal and malignant brain tissue (*Latikka & Eskola, 2019*), any

changes in the tumor size and composition will lead to changes in the impedance measured across the black box circuit network that is the head. Therefore, we suggest that monitoring changes in the electrical impedance between the TTFields delivery electrodes could be used to detect changes in the tumor treated by these electric fields, throughout the treatment, in real time. Once changes are detected, conventional medical imaging can be used to investigate the nature of these changes. This could replace randomly chosen times for medical imaging follow-up with follow ups that are clinically significant. Furthermore, if no changes in the impedance between the TTFields electrodes are detected, it may be an indication that the treatment is ineffective and that the treatment parameters should be modified. It should be emphasized, that measuring changes in electrical impedance between electrodes to monitor changes in the composition of tissue between the electrodes is not new. In fact, it serves as the basis for imaging techniques known as "electrical impedance tomography" (EIT) (*Escobar, 2020*) and magnetic induction tomography (MIT) (*Griffiths, 2001*). A closely related use of the concept described in this paper is to clinically monitor internal bleeding in the brain (*Gonzalez et al., 2013*), and to monitor cerebrovascular autoregulation (*Oziel et al., 2016*).

This paper is a first order feasibility experimental study, whose goal is to asses if measuring changes in electrical impedance between TTFields electrodes is sensitive enough to detect changes in the tumor size. In this study we explore the feasibility of the concept with a first order experimental model that employs a geometrically accurate head filled with a gel with electrical properties of brain tissue and in which the tumor is simulated with a tuber, with electrical properties of a tumor. The TTFields electrodes are attached to the skull and the experiments measure the changes in electric impedance between the TTFields electrodes as a function of the simulated tumor size and location. The other anatomic details of the head are considered fixed throughout the treatment and represented by the black box circuit network in Fig. 1B. This model is a first order approximation of human and tumor anatomy and is used to evaluate the feasibility of the concept.

We should also note that GBMS are not simply solitary masses but rather are often deeply infiltrating tumors, with scattered neoplastic cells extending out far from any observable mass on MRI. Moreover, the brain has ventricles which can take up a substantial volume of the cranial vault. However, the intent of the technique introduced in this study is to evaluate the feasibility of detecting changes in the main tumor mass from TTFields treatment by monitoring changes in impedance between the TTFields electrodes at a given frequency and for a fixed location of the TTField electrodes relative to the tumor. It should be emphasized that conventional monitoring of the effect of TTFields treatment is also done by monitoring the size of the tumor with MRI. The clinical efficiency of the treatment with TTFields and MRI means of treatment monitoring, has been demonstrated in thousands of treatment, worldwide.

While the brain and the tumor have a complex anatomy and composition, our measurements are geared to compare only the changes in impedance across the brain, before and after the TTFields treatment, with an eye towards the changes in the mass of the main tumor, as MRI does. The concept of using electromagnetic properties "changes" across the brain, to detect physiological changes in the brain, while the head is considered

as a black box, was validate in many clinical studies, *e.g.*, *Oziel et al. (2016)*, *Kellner et al. (2018)* and *Venkatasubba Rao et al. (2018)*.

## MATERIALS & METHODS

The study was performed on a geometrically accurate, 1:1 anatomically scaled human skull made of a durable polymer (Anatomy Warehouse, Evanston, IL, USA). The overall size is about 170 mm × 145 mm. The TTFields electrodes were placed on the skull in a way that precisely simulates clinical practice. Electrode arrays were attached to the skull at four locations, anterior, posterior, left and right, as shown in Fig. 1C. At each location there are 6 electrodes shorted together and connected to one measurement output. The 10 mm radius electrodes were designed to be similar to TTFields treatment electrodes (*Korshoej et al., 2017*). The TTFields simulating electrode arrays are made of a copper tape (Bertech, USA) with an insulated adhesive layer with which the electrodes are attached to the surface of the skull. The dimensions and placement of the electrodes on the skull and the output electric wire from each array are shown in Fig. 1C. The brain was simulated by a gel with electrical properties similar to those of bulk brain tissue (*Pomfret, Sillay & Miranpuri, 2013*; *Kandadai, Raymond & Shaw, 2012*). The gel was made of 4% alginate sodium + 2% NaCl + 94% deionized water. We have verified that this composition has electric properties similar to those of the brain in the frequency range of interest (test is not shown here). The GBM brain tumor, has a higher water content and an electrical conductivity that is several to ten times higher than the surrounding healthy tissue normal brain tissue (*Foster & Schwan, 1989*), which can provide sufficient sensitivity to evaluate the tumor change *via* impedance shift. The potato is used as a substitute for the tumor because the potato electrical conductivity at the frequency range of interest (*Gratz et al., 2021*) is similar to that of a tumor (*Latikka & Eskola, 2019*).

The potato was shaped into a sphere and placed at a predetermined location in the skull. Then, 800 ml gel was poured into the inverted skull, around the potato, as shown in Fig. 2B. The coordinates used for the placement of the potato (tumor) are shown in a schematic in Fig. 2C. The electrode arrays were connected to a precision Impedance Analyzer (Agilent 4294A) (Fig. 2A) to measure the impedance between each two opposite electrode arrays. Various dimensions of the potato sphere and their placement at various locations, were used to study the effects of tumor size and location on the impedance between the TTFields electrode arrays. Typically, TTFields are delivered in the frequency range of from 100 kHz to 300 kHz (*Fabian et al., 2019*), at which they produce the maximal therapeutic effect.

In this study, the impedance measurements were made at a frequency of 200 kHz, which is the preferred frequency used for clinical TTFields treatments. We report here only results from the change in amplitude measurements. We have shown in a series of previous studies that phase shift measurements at frequencies lower than 0.1 MHz cannot detect changes in tissue *e.g.*, *Gonzalez & Rubinsky (2006a)*, *González et al. (2006b)*, *Gonzalez, Horowitz & Rubinsky (2007)* and *González et al. (2009)*. Therefore, we report only experimentally measured changes in the impedance amplitude that resulted from the insertion of various tumor emulators in the skull, *i.e.*, $\Delta Z = |Z_2| - |Z_1|$.

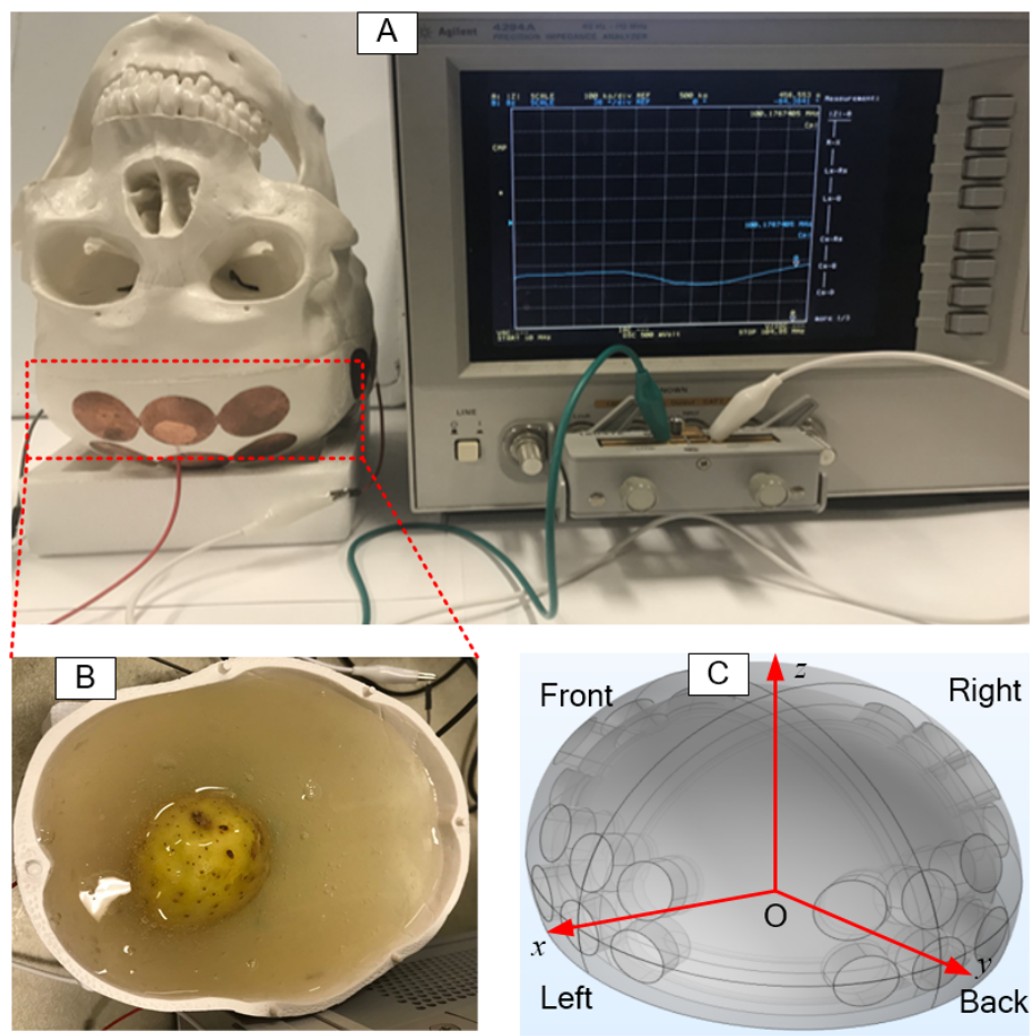

**Figure 2  Setup of experimental skull.** The setup of electrical impedance detection experiment on *in vitro* skull model, (A) skull model, (B) simulated brain tissue (gel) and tumor (potato), (C) the coordinate system of the model.

## RESULTS

In clinical practice, the TTFields electrode arrays are placed rigidly on the scalp at locations predetermined by the NovoTAL System (NovoTAL, USA), in such a way as to optimize the optimal delivery of the electric fields to the treated tumor. The treatment electrode arrays are usually placed in two opposite configurations, orthogonal to each other, which is considered to be the optimal layout for delivering the TTFields dose to the tumor (*Korshoej et al., 2017*). Therefore, the impedance between TTFields electrode arrays can be measured in two configurations, opposite and adjacent. Previous studies on placement of electrical impedance tomography (EIT) electrodes have shown that opposite electrodes measurements are more sensitive than adjacent electrode measurements

(*Koksal & Eyuboglu, 1995*). Therefore, our experiments were carried out with measurements made between opposite electrode arrays.

To establish a base line, the impedance between electrodes with the brain tissue phantom (gel) only, *i.e.,* without the tumor, was measured first. At a frequency of 200 kHz, the values of the impedance between the electrode arrays on the anatomical skull, filled only with the gel, without the simulated tumors, along the $x$-axis and the $y$-axis, are 372.25 $\Omega$ and 606.77 $\Omega$, respectively, this measurement was followed by experiments in which we inserted the tumor potato model at eight different locations on the $x$-axis and $y$-axis, in the simulating brain tissue gel. In these experiments the impedance was measured between the two opposite TTFields electrode arrays, at 200 kHz. Four repeated measurements were made for each experimental case. Figure 3A gives the experimental results showing the effect of tumor size and location on the changes in impedance relative to the baseline measurements. The maximal positive and negative deviations relative to the mean value were calculated from four repeat and are marked as error bars on the data points. It is evident that the change in impedance increases with the radius of the tumor. The figures show that it is possible to distinguish between the data point for every five mm increase in consecutive tumor radiuses, with a statistical significant difference of $F = 39.81 >$ F_crit $= 3.49$ and $P = 1.63 \times 10^{-6} < 0.05$, analyzed by ANOVA in Excel.

## DISCUSSION

The experimental results demonstrate that changes in tumor size can be monitored with typical TTFields electrodes and used to provide continuous information on the effects of the treatment on the tumor dimensions, with a resolution of at least five mm. The experimental data can be also used used to assess the sensitivity of the experimental impedance change measurements to the initial tumor radius. The first order derivative is a local measure of how changes in the abscissa affect changes in the ordinate. We performed this sensitivity analysis for two extreme conditions. Case A which deals with tumors at $x = 0$ which is the furthest location from the electrodes. Case B deals with tumors on $y = 40$, which is the proximate location to the electrodes. To calculate the derivative continuously, as a function of initial tumor radius, we used a cubic interpolation of the experimental points. The plots of the derivatives as a function of initial tumor radius are shown Fig. 3B, for cases A and B. It is evident that the sensitivity (first order derivative) increases with an increase in the initial radius of the phantom tumors in an exponential way and is larger for tumors that are close to the electrodes. This study shows that this technology is more effective at assessing the success of the TTFields treatment on larger tumors. This has clinical value, because the larger the tumor the larger the peril to the patient and the less time there is to optimize the treatment parameters.

In any real measurement system, the measurement noise is unavoidable. This section will discuss the effect of measurement noise on the accuracy. The conventional definition, of the Signal to Noise Ratio (SNR) (*Welvaert & Rosseel, 2013*) is:

$$\text{SNR} = 20\lg\frac{Z_s}{Z_n}. \tag{1}$$

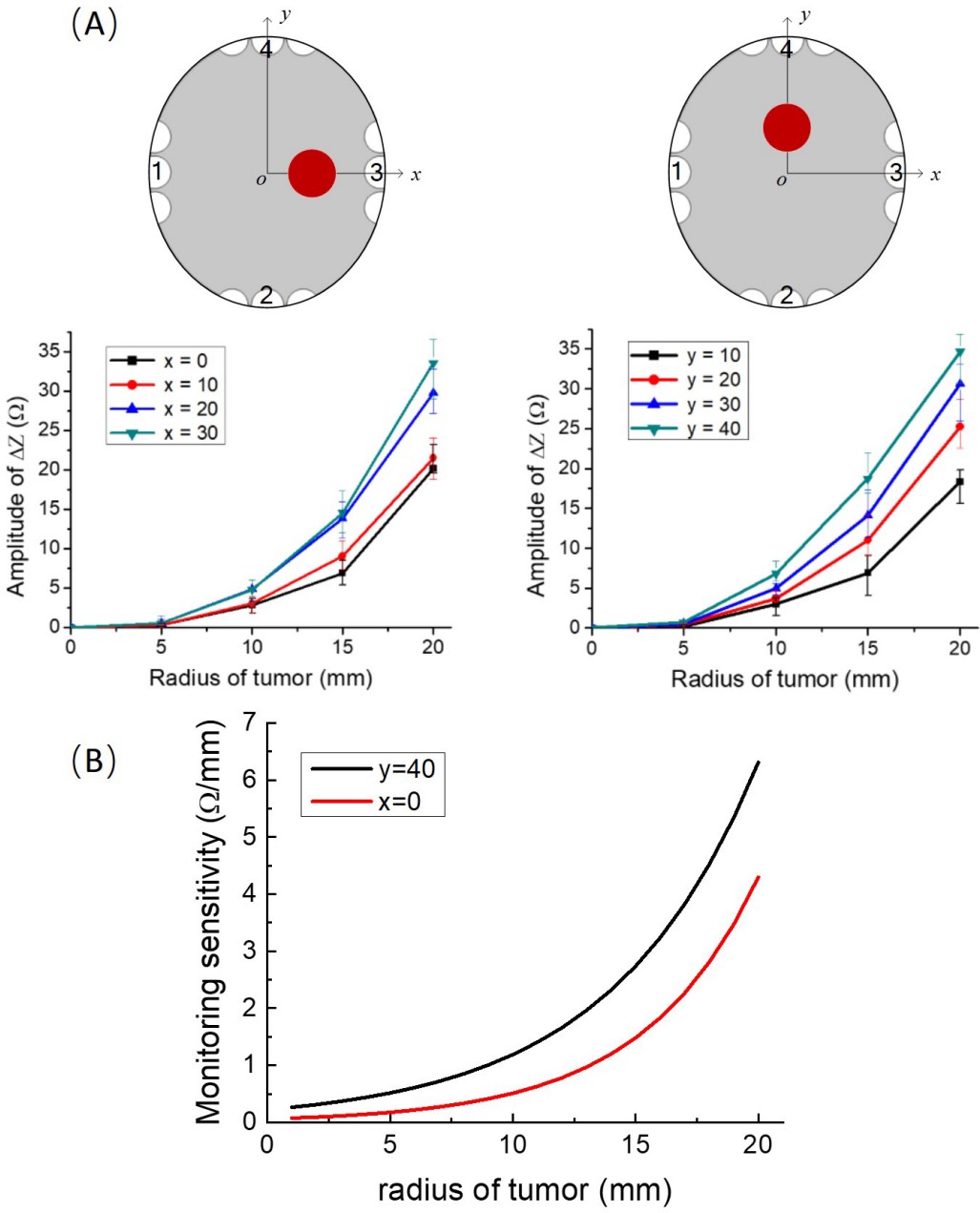

**Figure 3   Impedance as a function of tumor radius.** (A) Upper left row: schematics of locations of the tumor phantom. Lower left row: experimentally determined impedance change relative to the brain with no tumor, as a function of tumor size at different tumor locations on the *x* axis and *y* axis, error bars are shown for each data point. (B) The sensitivity to tumor changes. The first order derivative of the results in A.

where, $Z_s$ is the accurate measured impedance, $Z_n$ is the measurement impedance noise.

Considering the randomness of multiple measurements, the noise could be positive or negative. In this analysis, we used a random number generator to simulate the measurement noise, that is

$$Z_n = Z_s \cdot 10^{-\text{SNR}/20}(2 \cdot rand - 1). \tag{2}$$

where, the *rand* is a function that generates a random number between 0 and 1.

In the presence of noise, the impedance measured across TTFields electrode arrays will be

$$Z = Z_s + Z_n. \tag{3}$$

To ensure that the impedance change is not obscured by noise, the impedance change due to a change in tumor size must be larger than the measurement noise, that is

$$\Delta Z > Z_n = Z_s \cdot 10^{-\text{SNR}/20}(2 \cdot rand - 1). \tag{4}$$

As discussed earlier the baseline value of the measured impedance in the absence of tumors is between 372 Ω and 605 Ω. We perform here a calculation for a roughly median impedance measurement of 500 Ω. The highest resolutions of the impedance change for different noise levels was calculated from Eq. (4) and results show that, as expected, the resolution is improved with an increase in SNR. When the SNR is from 40 dB to 60 dB, the resolution of impedance changes with the tumor size between 5 Ω/mm and 0.5 Ω/mm. This implies that changes in tumor radius of one mm can be reliable detected only if it yields a larger than 0.5 Ω change in impedance. Our experimental data analysis in Fig. 3B, shows that this noise level is too high to detect one mm changes in tumor radius, even for a tumor as large as 20 mm. At 20 mm initial tumor radius, the first derivative is less than or close to 5 Ω/mm. However a SNR of 50 dB and higher is quite acceptable, in particular for larger tumors. Figure 3B shows that the first derivative for a tumor larger than 10 mm is about 1.21 Ω/mm ($y = 40$ mm, black curve in Fig. 3B) and 0.51 Ω/mm ($x = 0$, red curve in Fig. 3B). This result suggests that in practice, the SNR of the impedance measurement system should be higher than 50 dB. This is a reasonable level. The accuracy of the commercial Impedance Analyzer Agilent 4294A that was used in the experimental part of this study is 62 dB ($\pm 0.08\%$, data from the instrument manual). Higher precision electronics can be used to ensure an even higher SNR, for example precision electric bridges.

As for the simulated skull model, obviously, it is not an exact replica of the real human head and the real GBM tumor. In reality, the brain tissue is inhomogeneous with ventricles and the GBM tumor can have extensions in the brain. However, the materials used in the experiments have similar electric properties as the brain and the tumor. Therefore, considering the head a black box with constant electric properties, except for the change in size and location of the simulated tumor, can provide a qualitative assessment in a first order model examination of the effect of tumor changes on impedance across TTFields electrodes.

## CONCLUSION

TTFields are a relative new tissue ablation technology that treats cancer by affecting the division process in cancer cells. Treatment is done over periods of months and years and there is no simple technology to monitor the effectiveness of the treatment over time. Here we examined the idea that measuring the change in impedance across the electrode arrays that deliver the TTFields treatment could be used to monitor in real time the temporal changes in the tumor size. An *in vitro* skull experimental study has confirmed the potential value of the proposed monitoring system. Preliminary data suggests that measuring the change in impedance amplitude between opposite TTFields electrode arrays, at the frequency of the typical TTFields treatment parameters, can detect changes in the tumor radius with a resolution that increases with an increase in the initial radius of the tumor. This technique can be easily implemented by adding an impedance measurement function to the original commercial TTFields treatment device. The electrode arrays on the scalp can be designed to serve as a means to deliver the TTFields and as a sensor to measure the electrical impedance, between them. Obviously this is a first stage feasibility study that needs to be verified with clinical studies. If successful, this monitoring system may be a valuable addition to the TTFields cancer treatment technology. It is possible that in future implementations of this technique, an MRI image based exact simulation of the brain and the tumor, such as that generated by the NovoTAL System (*Trusheim et al., 2017*) (NovoTAL, USA) could be used to fine tune the interpretation of the measurements of the change in impedance between the TTFields electrodes.

While the brain and the tumor have a complex anatomy and composition, our measurements are geared to compare only the changes in impedance across the brain, before and after the TTFields treatment, with an eye towards the changes in the mass of the main tumor, as MRI does. It should be emphasized that the technique described in this study is designed for monitoring the effect of the TTFields procedure on the tumor, if it grows or recedes. The technique is not designed as a means to detect tumors. It is used only after the tumors are detected with other techniques that are much more precise, such as MRI. There is no doubt that MRI is much more sensitive and provides much more information than the technique in this paper. The advantage of this technique over MRI is that it is much less expensive, can be done at the home of the patient and continuously monitors the success of the treatment. However, we anticipate that this technique will be used primarily as a means to alert the physician that an MRI is immediately needed because the tumor seems to keep growing despite the treatment and that perhaps changes in treatment parameters are needed. Currently, the MRI's are scheduled at the physician chosen time intervals without any relation to the success of the treatment.

### Funding

The authors recieved no funding for this work.

## Competing Interests

Boris Rubinsky is an Academic Editor for PeerJ. The other co-authors declare that they have no competing interests.

## Author Contributions

- Xing Li conceived and designed the experiments, performed the experiments, analyzed the data, prepared figures and/or tables, authored or reviewed drafts of the paper, and approved the final draft.
- Moshe Oziel conceived and designed the experiments, authored or reviewed drafts of the paper, and approved the final draft.
- Boris Rubinsky conceived and designed the experiments, analyzed the data, prepared figures and/or tables, authored or reviewed drafts of the paper, and approved the final draft.

## Data Availability

The raw data for Fig. 3 is available in the Supplementary File.

## Supplemental Information

Supplemental information for this article can be found online at http://dx.doi.org/10.7717/peerj.12877#supplemental-information.

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
