# Peer review of "Evaluating the therapeutic effect of tumor treating fields (TTFields) by monitoring the impedance across TTFields electrode arrays"

_PeerJ, doi:10.7717/peerj.12877_

## Round 0.1 · original submission · Major Revisions

Please address the concerns of both reviewers and amend your manuscript accordingly.

Reviewer 1 ·

Basic reporting

The article is written clearly, with the introduction providing motivation for the problem addressed. TTF is a therapeutic modality that gaining wider interest, and having a real-time reporting/monitoring system would be beneficial. The literature and background/context for the proposed solution of EIT is well motivated. The article structure, figures and tables as presented are acceptable.

Experimental design

The authors present an original result where the use a head phantom to test whether electrode configurations used for TTF can detect a tumor mass via impedance measurements. The results are in align with the hypothesis and demonstrates the feasibility of the approach.

Validity of the findings

The only perceivable weakness of the topic is that it reports on the feasibility of impedance measurements using TTF electrode arrays, that would be further strengthened if actual treatment vs. response were to be measured. Perhaps these experiments could be performed in a petri dish setting.

Further, the use of a potato, which is a heterogenous mass, in an otherwise homogeneous background is expected to provide distinct results as seen here. More nuanced studies or measurements where the inclusion has much less dramatic conductivity differences in comparison to the background would be of interest.

Reviewer 2 ·

Basic reporting

This is a fairly basic paper evaluating the potential of applying electrical impedance using tumor treating field electrodes as a means of assessing tumor response to therapy. In other words, one set of electrodes could serve as both a means for the treating the tumor and measuring the effect of treatment. The paper uses a fairly basic model of a skull, a potatoes of different sizes to mimic the tumor and a gel to mimic brain tissue. The demonstrate that impedance changes can be detected and these mirror tumor size. A sensitivity analysis is also completed.

The language is clear, the figures, sufficient, references look complete.

But the article is incredibly sparse and many details are omitted (see below).

Table 1 does not need to be a table.

Experimental design

The experimental design is very straightforward, but the model is a very weak approximation of human anatomy and the tumor behavior. GBMS are not simply solitary masses but rather are often deeply infiltrating tumors, with scattered neoplastic cells extending out far from any observable mass on MRI. Moreover, the brain has ventricles which can take up a substantial volume of the cranial vault. It is entirely unclear how the non-homogeneities in normal brain would impact this model as well. Thus the model is really quite unrealistic, yet there is absolutely no discussion as to these huge limitations.

Moreover there is often considerable edema associated with these tumors...again another limitation that this model does not include.

We are told very little about the impedance methods used except the frequency (200 kHz), the rationale of which is not provided. It is also unclear which electrodes were used for the impedance measurements, and were these 4-electrode measurements (I think so)? And if so, which are the voltage and which are the current electrodes.

What impedance system was used?

Only the impedance magnitude was measured. Why? They mention simulations showed that phase was less sensitive, but why not provide that information too?

Validity of the findings

The validity is limited as noted above based on the very simplistic nature of this model.

There is no inclusion of any kind of skin layer which could also impact the findings.

The conclusions are vastly overstated the limitations of this very simplistic model are not included.

Additional comments

See comments regarding limitations of the study design above.

Overall, the results decided unsurprising and I think this adds relatively little to the literature, although the basic idea of using electrical impedance along with TTFs is good. I really which the authors had spent more time constructing a more realistic model than that presented here. The article is also short on many details. While the results here could be useful for further developmental work, they could also be highly misleading.

---

## Round 0.2 · accepted · Accept

Thank you for addressing all the issues pointed by the reviewers and for amending the manuscript accordingly. I am pleased to accept the revised manuscript now.

Reviewer 1 ·

Basic reporting

Appropriate as presented.

Experimental design

Acceptable.

Validity of the findings

Acceptable.